



# Evolution of Organic Carbon in the Laboratory Oxidation of Biomass Burning Emissions

Kevin J. Nihill,[1,a] Matthew M. Coggon,[2,3] Christopher Y. Lim,[1,b] Abigail R. Koss,[2,3,4,c] Bin Yuan,[2,3,d] Jordan E. Krechmer,[3,4,e] Kanako Sekimoto,[2,3,5] Jose L. Jimenez,[3,4] Joost de Gouw,[2,3,4] Christopher D. Cappa,[6] Colette L. Heald,[1,7] Carsten Warneke,[2,3] Jesse H. Kroll[1,8]

[1] Department of Civil and Environmental Engineering, Massachusetts Institute of Technology, Cambridge, MA, USA

[2] NOAA Chemical Sciences Laboratory, Boulder, CO, USA

[3] Cooperative Institute for Research in Environmental Sciences, University of Colorado, Boulder, CO, USA

[4] Department of Chemistry, University of Colorado, Boulder, CO, USA

[5] Graduate School of Nanobioscience, Yokohama City University, Yokohama, Kanagawa, Japan

[6] Department of Civil and Environmental Engineering, University of California, Davis, CA, USA

[7] Department of Earth, Atmospheric and Planetary Sciences, Massachusetts Institute of Technology, Cambridge, MA 02139, USA

[8] Department of Chemical Engineering, Massachusetts Institute of Technology, Cambridge, MA 02139, USA

[a] now at: University of Chicago Laboratory Schools, Chicago, IL, USA

[b] now at: South Coast Air Quality Management District, Diamond Bar, CA 91765, USA

[c] now at: Tofwerk A.G., Boulder, CO, USA

[d] now at: Institute for Environmental and Climate Research, Jinan University, Guangzhou 511443, China

[e] now at: Bruker Scientific, Inc., Billerica, MA, USA

*Correspondence to:* Kevin Nihill (kevin.j.nihill@gmail.com), Jesse Kroll (jhkroll@mit.edu)

**Abstract.** Biomass burning (BB) is a major source of reactive organic carbon into the atmosphere. Once in the atmosphere, these organic BB emissions, in both the gas and particle phases, are subject to atmospheric oxidation, though the nature and impact of the chemical transformations are not currently well constrained. Here we describe experiments carried out as part of the FIREX FireLab campaign, in which smoke from the combustion of fuels typical of the Western U.S. was sampled into an environmental chamber and exposed to high concentrations of OH, to simulate the equivalent of up to two days of atmospheric oxidation. The evolution of the organic mixture was monitored using three real-time time-of-flight mass spectrometric instruments (a proton transfer reaction mass spectrometer, an iodide chemical ionization mass spectrometer, and an aerosol mass spectrometer), providing measurements of both individual species and ensemble properties of the mixture. The combined measurements from these instruments achieve a reasonable degree of carbon closure (within 15-35%), indicating that most of the reactive organic carbon is measured by these instruments. Consistent with our previous studies of the oxidation of





individual organic species, atmospheric oxidation of the complex organic mixture leads to the formation of species that on average are smaller and more oxidized than those in the unoxidized emissions. In addition, comparison of mass spectra from
the different fuels indicates that the oxidative evolution of BB emissions proceeds largely independent of fuel type, with different fresh smoke mixtures ultimately converging into a common, aged distribution of gas-phase compounds. This distribution is characterized by high concentrations of several small, volatile oxygenates, formed from fragmentation reactions, as well as a complex pool of many minor oxidized species and secondary organic aerosol, likely formed via functionalization processes.

## 1 Introduction

Biomass burning (BB) is a major source of reactive organic carbon, both in the form of aerosol and non-methane organic gases (NMOG), to the atmosphere (Carter et al., 2022; Andreae et al., 1988; Yokelson et al., 2009; Akagi et al., 2013; Bond et al., 2004; Liu et al., 2016). BB emissions can have large impacts on air quality and climate across a range of scales, due both to primary species and to their transformation to secondary species such as ozone and secondary organic aerosol
(SOA) (Cubison et al., 2011; Decarlo et al., 2010; Grieshop et al., 2009; Hennigan et al., 2011; Ortega et al., 2013; Tkacik et al., 2017; Yokelson et al., 2009; Hobbs et al., 2003; Akagi et al., 2013; Bourgeois et al., 2021; Hobbs et al., 1997; Liu et al., 2014). Primary and secondary BB species also include species that are known to be hazardous to human and ecosystem health (Künzli et al., 2006; Naeher et al., 2007; O'Dell et al., 2020).

Whereas many studies have measured the composition of primary BB emissions, the formation and evolution of
secondary species are less well characterized. Those studies that have examined secondary chemistry have focused largely on the formation and properties of SOA (Lim et al., 2019; Ortega et al., 2013; Cubison et al., 2011) or individual species and pathways in BB photooxidation (Coggon et al., 2019; Koss et al., 2018; Sekimoto et al., 2018). These laboratory-based studies have revealed that atmospheric processing of BB emissions can lead to multiple generations of product formation as well as SOA formation. SOA formation within BB plumes has been largely attributed to the presence of furans and oxygenated
aromatic compounds (Akherati et al., 2020), which are not particularly abundant but comprise a sizeable fraction of the overall OH reactivity.

Despite these recent process-based studies uncovering key SOA precursors, the overall evolution of the entire reaction mixture – how the organic carbon in both the gas and particle phases evolves as a whole (Heald and Kroll, 2020) – has received comparatively less attention. Characterization of the changes to chemical composition and properties of BB emissions is
important for understanding and predicting their impacts (e.g., on photochemical cycling, ozone production, and aerosol loadings and properties) over a wide range of temporal and spatial scales. Additionally, such studies may aid in the identification of aged BB plumes in field studies, as well as in the development of improved oxidation mechanisms of BB mixtures for use in chemical transport models.



Here we examine the oxidative evolution of the entire organic mixture within BB smoke, by carrying out laboratory
studies that simulate atmospheric oxidation of the emissions, and monitoring their evolving composition using multiple
analytical instruments. In recent work, we have demonstrated the ability of a suite of mass spectrometric instruments to
comprehensively measure a large fraction of the total carbon in a given oxidation system (Isaacman-VanWertz et al., 2018,
2017; Koss et al., 2020). Whereas those studies focused primarily on single precursor molecules, which then react to form a
wide range of products, here we start with an already-complex mixture of organic species, and examine its further evolution
upon oxidation. Specifically, as part of the FIREX FireLab 2016 campaign, a range of fuels were burned and sampled into a
reactor where they were oxidized over several atmosphere-equivalent days and characterized using three mass spectrometric
instruments. By characterizing the changing properties of BB emissions on both an ensemble and a per-molecule basis, we are
able to identify trends and commonalities in the oxidation of these complex mixtures; this provides a more complete picture
of the evolving product distributions and an improved understanding of how emissions from different fuels are processed in
the atmosphere.

## 2 Materials & Methods

### 2.1 Sampled Burns

Laboratory experiments were conducted as part of the Fire Influence on Regional and Global Environments Experiment
(FIREX) FireLab 2016 campaign, at the US Forest Service Fire Sciences Laboratory in Missoula, MT, USA in October 2016.
The broad aim of this campaign was to better understand biomass burning emissions and atmospheric aging, with a focus on
fuels characteristic of the Western United States.

Detailed descriptions of the campaign, experimental setup, list of fuels burned, and resulting emissions of key
NMOGs are provided elsewhere (Koss et al., 2018; Lim et al., 2019; Coggon et al., 2019). The subset of experiments discussed
here was carried out as part of the "stack burn" component of the campaign. A known amount (350-3000 g) of a given fuel
was burned underneath the central exhaust stack. Here, we focus on the subset of burns for which data is available from all
three analytical instruments used in this study (discussed below); these burns and their corresponding fuels are listed in **Table
1**. This list includes one replicate (Lodgepole Pine Litter) and a comparison of sub-types of fuel (Engelmann Spruce canopy
vs. duff); it also includes one blank run (in which Lodgepole Pine smoke was sampled but not oxidized).

| Fire No. | Fuel | Type | MCE* | Moisture content (%) |
|---|---|---|---|---|
| 21 | Lodgepole Pine | Litter | 0.9255 | 11.7 |
| 25 | Engelmann Spruce | Canopy | 0.9499 | 34.0 |



| 26 | Engelmann Spruce | Duff | 0.8171 | 0.6 |
| 38 | Ponderosa Pine | Litter | 0.9448 | 6.2 |
| 41 | Lodgepole Pine | Litter | 0.9377 | 7.5 |
| 63** | Lodgepole Pine – no UV | Mix | 0.9364 | 38.5 |

Table 1. List of fuels whose emissions were sampled into the mini-chamber for oxidative aging in this work. *MCE = Modified Combustion Efficiency, equal to $[CO_2]/([CO_2]+[CO])$. **I-CIMS data were not available for this experiment.

## 2.2 Mini-chamber

All aging experiments were carried out in the "mini-chamber," a 150 L perfluoroalkoxy (PFA) environmental chamber (Lim et al., 2019). Prior to each experiment, the chamber was flushed with clean, humidified air for ~45 min. For each burn studied, smoke from the top of the stack was transported to the wind tunnel room in the Fire Sciences Laboratory (FSL) via a 30 m long, large-diameter community inlet and a fast flow rate to minimize interactions with walls of the inlet. Smoke from this inlet was then sampled from the center of the flow using an ejector diluter pressurized with clean air and passed through 1 m of stainless-steel tubing and a $PM_1$ cyclone before being injected into the chamber at a dilution of 1:10 with clean air. Filling continued until particle concentrations reached a maximum. Prior to oxidation, 40 ppb of deuterated butanol (an OH tracer) was injected into the chamber, followed by a constant stream of clean, humidified (RH ~ 30%) air doped with 50-100 ppb ozone, with a flow to match instrument sampling flows. After allowing time for mixing, one 40 W UVC lamp (narrow peak emission at 254 nm, Ultra-Violet Products, Inc.) was turned on to generate OH (via photolysis of ozone, followed by reaction of $O(^1D)$ with $H_2O$, as well as the photolysis of other precursors in the smoke such as HONO) which initiated oxidation. Photochemistry proceeded for ~30-45 min, during which time the chamber was constantly diluted with room-temperature, humidified air (RH ~30%) doped with ~70 ppb $O_3$ as the contents were sampled by different instruments ("semi-batch mode"). While UVC irradiation is capable of introducing other photochemical (i.e., non-OH) reactions, OH oxidation has been previously shown to be the dominant oxidation mechanism in this type of system (Peng et al., 2016; Coggon et al., 2019). In one experiment (Fire 63; see **Table 1**), the UV lights remained off, ensuring OH was at near-zero levels and providing a blank for comparison with the other runs.

$RO_2$ chemistry within the mini-chamber differs somewhat from that of typical atmospheric plume conditions, with $RO_2+NO_2$ and $RO_2+RO_2$ reactions generally more important than in the atmosphere (Coggon et al., 2019). However, over the course of a given aging experiment, the chemistry exhibits a shift from high-NO ($RO_2+NO$) to low-NO ($RO_2+HO_2$) conditions, similar to what occurs in atmospheric plumes. The evolution of these chemical regimes depends partially on the availability of $NO_x$, which is higher for fires characterized by high MCE emissions (Roberts et al., 2020). For fires with very low $NO_x$ emissions (low MCE), the transition to low-$NO_x$ chemistry can occur within short distances downwind of fires (Xu et al., 2021).

## 2.3 Analytical Techniques



Several analytical instruments were run to monitor the evolving composition of the smoke in real time. Detailed descriptions of these are provided elsewhere (Koss et al., 2018; Lim et al., 2019; Coggon et al., 2019), but a brief overview is included here. Particle-phase composition was measured by a high-resolution time-of-flight aerosol mass spectrometer (AMS, Aerodyne, Inc.) that collected data at 1-minute intervals. While the hard ionization technique employed by this instrument precludes the identification of individual molecules, it enables the determination of H:C and O:C elemental ratios and the mean oxidation state of particle-phase carbon. Corrections for wall loss, dilution, and collection efficiency were carried out individually, as described in *Lim et al.* (2019) The overall absolute uncertainty in measurement of organic particulate mass is estimated to be 38%, as discussed elsewhere (Bahreini et al., 2009).

Measurements of the gas phase were made by two real-time time-of-flight chemical ionization mass spectrometers, a custom proton-transfer-reaction mass spectrometer (PTR-MS, described elsewhere (Yuan et al., 2016)) and an iodide-cluster chemical ionization mass spectrometer (I-CIMS, Aerodyne/Tofwerk, AG (Lee et al., 2014)), both measuring at one-second time resolution. The PTR-MS was used to detect and quantify the less-oxidized compounds (e.g., hydrocarbons and simple oxygenates); operation and data analysis are described elsewhere (Koss et al., 2018; Coggon et al., 2019). Many of the species detected by the PTR during FIREX were subsequently identified via a combination of inspecting formulae, mapping out likely reaction pathways, and considering isomer-dependent volatilities determined via in-line gas chromatography-mass spectrometry (GC-MS) (Koss et al., 2018). Of the 291 unique ions detected by the PTR, 134 were identified, and the remaining and 157 are unidentified (i.e., are represented by their molecular formulae alone). Instrument signals (in normalized counts per second, ncps) were then converted into gas-phase concentrations (ppbv) using either direct calibrations (when available) or approximate calibration methods based on proton-transfer-reaction rate constants (Koss et al., 2018). Uncertainties in concentrations were determined based on calibration uncertainties: 15-50% for identified species and a factor of 2 for unidentified ones (Sekimoto et al., 2017).

The I-CIMS was used for the measurement of highly oxidized, multifunctional (often lower-volatility) compounds. Details of its use in FIREX, including relevant instrument parameters, are described elsewhere (Coggon et al., 2019). The I-CIMS measured 522 unique iodide-cluster ions (ions without an I in the formula were not considered). Due to unavailability of standards, I-CIMS signals could not be meaningfully calibrated into concentrations for the vast majority of ions; only a small subset of ions was calibrated directly, allowing for the determination of a maximum instrument sensitivity (~8000 ncps/ppt). All species measured by the I-CIMS were assigned this calibration factor. Concentrations were assigned a factor of 2.5 uncertainty due to a lack of calibration standards (Isaacman-VanWertz et al., 2018; Robinson et al., 2022).

While the PTR and I-CIMS largely measure different molecules comprising the reaction mixture, there are several instances in which both instruments detect identical elemental formulae. Potential double-counting of such compounds is avoided by comparing the time series of isomeric species to identify those compounds that were detected by both instruments. In the case of a "match" between two instruments ($R^2 > 0.8$ in terms of the temporal behavior in a single experiment), the I-CIMS trace was discarded due to its greater uncertainty relative to the PTR measurement. Overall, 50 overlapping traces were removed, leading to a total of 763 unique gas-phase species.





The identification of molecular formulae (and therefore elemental composition) for these gaseous compounds enables the calculation of several physicochemical properties, such as carbon oxidation state ($\overline{OS_C}$), saturation vapor pressure ($c*$) and reaction rate constant with OH ($k_{OH}$). For identified species, the measured/exact values are used when available (Koss et al., 2018). For all other (unidentified) species and when measured/exact values are not available, these are instead estimated using structure-activity relationships (Daumit et al., 2013). $\overline{OS_C}$ values are calculated following conventions detailed in (Kroll et al.,

2011). This approach assumes that all O atoms have an oxidation state of –2 (i.e., that they are not peroxides), and that organic N is in the form of nitrooxy groups (oxidation state of +5) if there are at least 3 O atoms or amines (oxidation state of –3) if there are not.

      Certain classes of compounds were not directly measured due to experimental limitations. Concentrations of CO and $CO_2$ formed in the oxidized mixture are very small compared to the high levels of these species emitted from the combustion

itself and are therefore not included in this analysis. A high background signal for butanol (possibly due to a leak from a neighboring instrument) precluded the precise contribution of this compound to the product distribution. Peroxyacyl nitrate (PAN) compounds are likely to be present in many of the reaction mixtures (Coggon et al., 2019; Müller et al., 2016), but these compounds cannot be unambiguously measured by the PTR (Hastie et al., 2010; Hansel and Wisthaler, 2000). While these species are likely to be accounted for among the unidentified compounds, their exact concentrations in the reaction mixture

are poorly constrained.

      Dilution of gas-phase species was corrected for by monitoring the decay of acetonitrile, which is present at high concentrations in BB smoke, is slow to react with OH ($k_{OH} = 2.16\times10^{-14}$ cm$^3$ molec$^{-1}$ s$^{-1}$), and is not substantially lost to Teflon surfaces (Krechmer et al., 2016; Pagonis et al., 2017). The instrument flow rates and volume of the mini-chamber resulted in a physical residence time on the order of tens of minutes, corresponding to several days of atmospheric-equivalent aging by

OH (determined from the decay of deuterated butanol, and assuming an ambient OH concentration of $1.5\times10^6$ molecules/cm$^3$). At reaction times beyond ~20 minutes (~2 days of atmospheric oxidation), the high dilution leads to an increased likelihood of evaporation of gaseous compounds from chamber walls and SOA particles, and the background levels become an increasing fraction of the total dilution-corrected signal. Thus, measured concentrations beyond such reaction times are unreliable, and we limit all experiments to just these first 2 days of atmospheric-equivalent timescales.

## 3 Results & Discussion

      For all burns studied in this work, the reaction mixture changed dramatically as a function of oxidation, with most of

the initially present species reacting away and many other species growing in. This is shown in Fig. 1, which presents stacked plots of the changing concentrations of all gas- and particle-phase organics in the oxidation of smoke from two burns (Fire 25 / Engelmann Spruce canopy and Fire 26 / Engelmann Spruce duff) over two days of atmosphere-equivalent aging. (Equivalent plots for the other burns are in Fig. S1.) Traces are colored by the instrument used for detection/quantification, and gas-phase





species are separated further into bands representing compounds that are primarily consumed (dark red/blue) or formed (light
red/blue) throughout the course of oxidation. Comparison of these burns with the blank (Fire 63, Lodgepole Pine with no UV;
see Fig. S1d), in which little change in composition is observed, further demonstrates the extent to which these reaction
mixtures are oxidatively aged. Error bars represent the uncertainty (1 ) of the total concentrations before and after aging, for
individual instruments as well for the overall reaction mixture. The relatively small uncertainty is attributable to both the low
uncertainty in PTR measurements (due to many compounds being detected to within 15-50%, as discussed above) and the
uncorrelated uncertainties in measurements that result in a small total fractional uncertainty after adding individual errors in
quadrature. Throughout the course of the experiments, the "unidentified" fraction of the gas-phase mixture was observed to
increase substantially: they made up 12%-23% of the measured gas-phase organic carbon in the fresh smoke, but this fraction
increases to 22%-29% after oxidation. The higher uncertainties in the calibration factors for these species leads to an increase
in overall uncertainty in total organic carbon at the end of a given experiment.

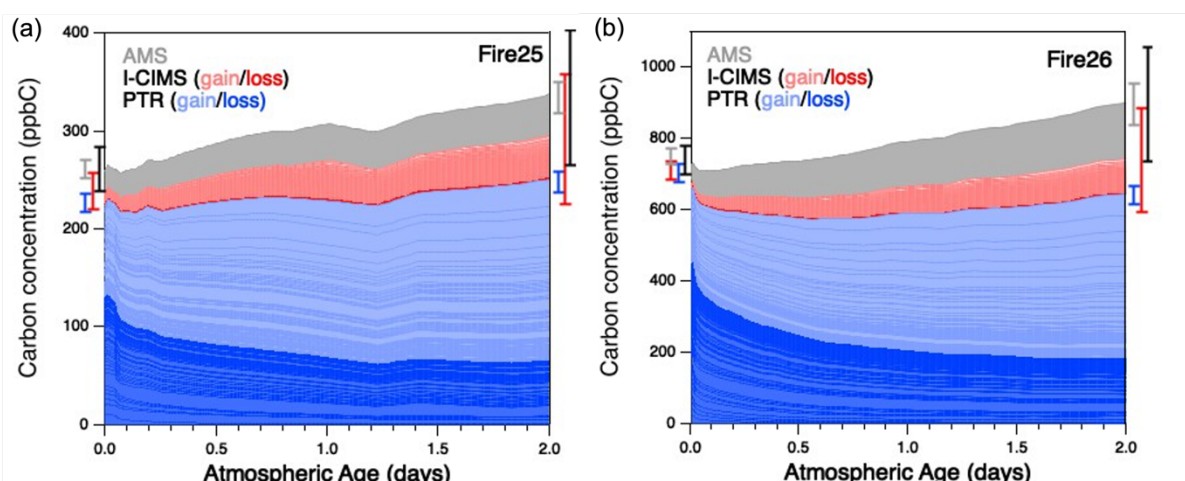

Figure 1. Measured organic carbon in the oxidation of emissions, as a function of estimated oxidative age, from (a) Fire 25 /
Engelmann Spruce canopy and (b) Fire 26 / Engelmann Spruce duff, as a function of atmosphere-equivalent age. Measurements
are separated into individual bands according to the instrument by which they were detected. Gas-phase measurements are
separated further: blue traces represent species measured by the PTR that are primarily consumed (dark) or formed (light),
ranked in order of largest decay (bottom) to largest increase (top); red traces follow the same convention, but for the I-CIMS.
The gray trace represents OA measurements made by the AMS (Lim et al., 2019). The uncertainty (1σ, denoting calibration
uncertainties only) for each instrument is shown to the left and right of the plot, corresponding to uncertainty before and after
atmospheric aging, with total uncertainty (black error bar) calculated by adding together individual uncertainties in quadrature.





In all cases (Figs. 1 and S1) there is a clear overall increase in total measured organic carbon over the course of the reaction; this increase is not large (typically 15-35%). While it falls within the total uncertainty (1σ) of the measurements, the fact that it is seen in all aging studies suggests that systematic errors (and not just random calibration errors) may be playing a role. First, some classes of species are unmeasured by our instrument suite. Neither the PTR nor I-CIMS can measure alkanes, which may be present at the beginning of each run (Koss et al., 2018), and cannot be formed photochemically. While the exclusion of alkanes would lead to an underestimate of starting carbon, their relatively low emission factors (e.g., 0.02-0.08 g kg$^{-1}$ for $n_C \geq 4$ species, (Andreae, 2019)) make them unlikely to account for more than ~5% of the "missing" carbon at the beginning of each run. On the other hand, as noted above, secondary CO and $CO_2$ production was not monitored either. In addition, as noted above, the increasing dilution of the reaction mixture may lead to changes in background concentrations and wall effects, including the partitioning of semivolatile species from the chamber walls or the oxidation of species deposited on the walls; these may result in an increase in measured carbon throughout the course of the reaction. Finally, wall loss of gas-phase species (both primary and secondary species), while unlikely to be exceedingly large (Lim et al., 2019), was not directly corrected for in the mini-chamber, and could affect total carbon balance. Together, these factors may lead to over- or under-estimation in total carbon in complex ways, but the net effect is an apparent increase in total measured carbon as the reaction mixture is aged.

Despite these uncertainties, the approximate carbon closure (to within 15-35%) in these experiments suggests that the majority of reactive organic carbon is accounted for. This enables the evolution of the reaction mixture as a whole to be examined. As an example of such mixture evolution, Fig. 2 shows the changes of two key ensemble characteristics of the gas-phase reaction mixture – mean carbon oxidation state ($\overline{OS_C}$) and carbon number ($n_C$) – for each individual burn as a function of atmospheric age. These measurements represent the entire gas-phase mixture, including both primary emissions and secondary organic reaction products. Particulate carbon is not included since $n_C$ (and other individual molecular parameters) cannot be measured by the AMS; given that the AMS contributes <20% of the overall carbon mass, its contribution to overall $\overline{OS_C}$ is likely to be modest. The trends for each fire are derived from the time-dependent distributions of $\overline{OS_C}$ and $n_C$ for each burn, shown for Fire 26 in Figs. 2c-d and for all other burns in Figs. S2-S3. All burns characterized here, independent of fuel, exhibit the same overall trends, in which OH oxidation leads to increases in the oxidation state of the organic carbon (Fig. 2a), as well as decreases in mean carbon number (Fig. 2b) owing to fragmentation reactions. These trends are in line with expectations for oxidative systems (Kroll et al., 2011) and are qualitatively consistent with those observed for the oxidative evolution of α-pinene upon reaction with OH (Isaacman-VanWertz et al., 2018). This is in contrast to the blank run (which involved no UV irradiation and hence negligible OH), for which properties of the reaction mixture remained essentially constant throughout the experiment. (Any minor changes that do occur in the blank run may result from differences in losses of gas-phase species to the chamber walls, or simply reflect the measurement uncertainty.) Distributions of other key chemical parameters, such as volatility ($c^*$) and oxidative lifetime ($\tau_{OH}$), are not measured directly, but can be estimated from speciated measurements or structure-activity relationships (Daumit et al., 2013). These trends are shown in Figs. S4-6: $c^*$ is observed to





experience a brief increase before gradually decreasing with further oxidation, while $\tau_{OH}$ is found to continually increase

throughout the course of oxidation. As with $\overline{OS_C}$ and $n_C$, trends are largely the same across all burns.

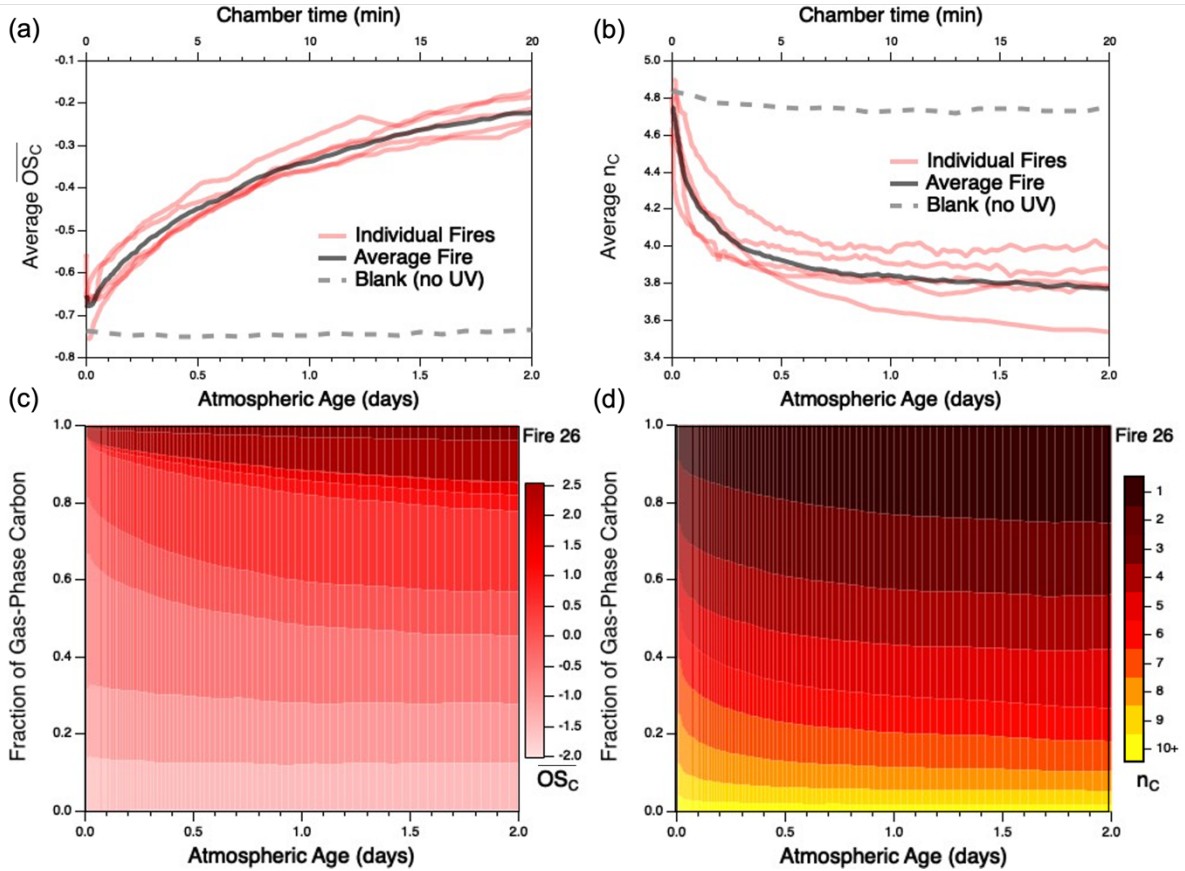

Figure 2. Evolution of (a) mean carbon oxidation state ($\overline{OS_C}$) and (b) mean carbon number ($n_C$) for the measured gas-phase

species, as a function of atmospheric aging (or chamber time for the blank experiment). Red lines represent individual fires;

gray dashed lines represent the blank run, for which only PTR data is used; black line denotes the average of all burns studied,

as described later in the text. Traces for individual fires are averages of the evolving distribution of $\overline{OS_C}$ and $n_C$ from each fire;

these are given for Fire 26 (Engelmann Spruce duff) in panels (c) and (d), respectively. Evolution of $\overline{OS_C}$ and $n_C$ distributions

for the remaining fires can be found in Figs. S2-3; corresponding figures for volatility ($c^*$) and oxidative lifetime ($\tau_{OH}$) are

given in Figs. S4-6.

Figure 3 shows the simultaneous evolution of both $\overline{OS_C}$ and $n_C$ distributions for different burns. All measured gas-

phase compounds from Fires 25 (Engelmann Spruce canopy; filled blue circles) and 26 (Engelmann Spruce duff; open red



circles) are displayed in $\overline{OS_C}$ vs. $n_C$ space, first for fresh emissions (Fig. 3a) and then after 2 atmosphere-equivalent days of oxidation (Fig. 3b), with circle area proportional to normalized carbon-weighted concentration (in ppbC). The loading and $\overline{OS_C}$ of the organic particulate matter are also shown (in green); however, since the AMS provides no information on carbon number, these are presented on a separate axis and at a different size scale than the gas-phase data (since the total particulate mass is much larger than that of any single gas-phase compound). To the top and right of each plot, histograms represent projections of gas-phase distributions of $n_C$ and $\overline{OS_C}$, respectively.

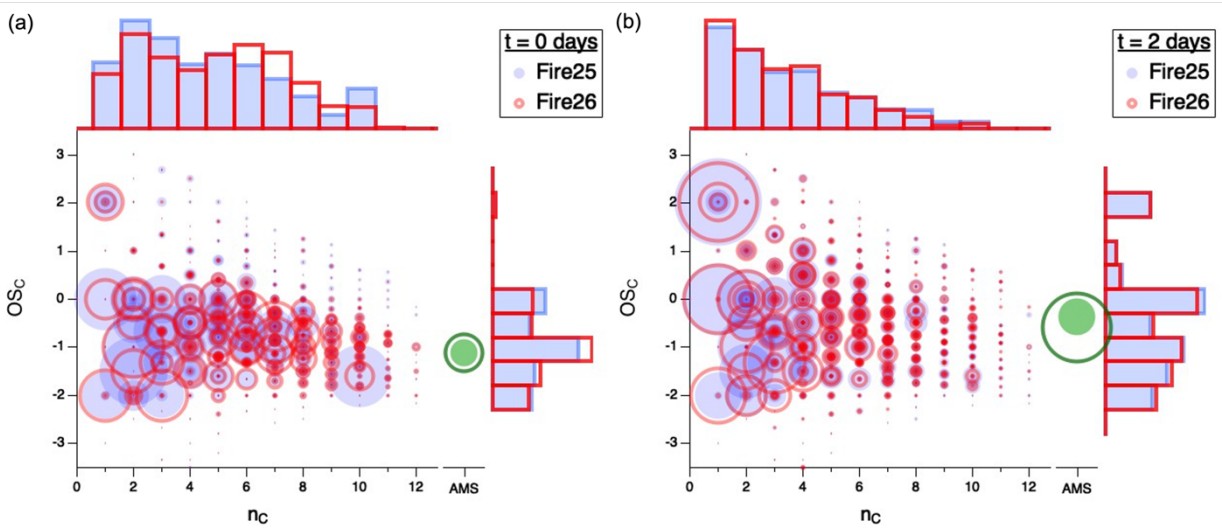

Figure 3. Comparison of carbon oxidation state ($\overline{OS_C}$) vs. carbon number ($n_C$) for gas-phase emissions from Fire 25 (Engelmann Spruce canopy; filled blue circles) and Fire 26 (Engelmann Spruce duff; open red circles) for (a) freshly sampled emissions and (b) after two days of atmosphere-equivalent oxidation. Area of each circle is proportional to concentration (ppbC), normalized to total carbon concentration for comparison between fuels. The separate green circles (filled for Fire 25 and open for Fire 26) denote particle-phase carbon, for which $n_C$ measurements are not available; the area scaling is different from the scaling of the gas-phase data. Histograms along the top and right axes show $\overline{OS_C}$ and $n_C$ distributions of the gas-phase products. Equivalent figures for the other fires, along with videos showing the change in product distributions over time, can be found in the SI.

For the fresh emissions (Fig. 3a), the organic carbon spans a wide range of $\overline{OS_C}$-$n_C$ space; carbon numbers vary from 1 to >10, with most species having mean carbon oxidation states between -2 and 0. After 2 days of atmosphere-equivalent oxidation (Fig. 3b), there is a marked shift towards smaller carbon numbers and higher oxidation states (consistent with Fig. 2), largely driven by the loss of large (high-$n_C$) species and the formation of small oxygenates, such as formic acid ($n_C = 1, \overline{OS_C} = 2$), acetic acid ($n_C = 1, \overline{OS_C} = 0$), and formaldehyde ($n_C = 1, \overline{OS_C} = 0$). The trend towards smaller species is exemplified in the carbon number histogram, which evolves from a reasonably broad and disordered distribution in the fresh smoke to one in which abundance decays nearly exponentially with $n_C$ in the aged mixtures. The oxidation state of the organic





particulate matter also increases, as described in detail in *Lim et al.* (2019). Specifically, that study showed that the average

increase in $\overline{OS_C}$ across various fuels is 1.33 ± 0.50, driven by continual oxidation of the gas-phase mixture, Leading to the

formation of lower-volatility organic species and hence SOA. The greater SOA formation (green circles) in Fire 26 likely

arises from the larger initial concentrations of VOCs, and especially the greater abundance of phenolic compounds (e.g.,

phenol, cresol, catechol, guaiacol), in that fire (Akherati et al., 2020; Lim et al., 2019). Notably, the $\overline{OS_C}$-$n_C$ distributions of

the two sets of aged emissions (Fig. 3b) are much more similar than those of the fresh emissions (Fig. 3a), suggesting a

convergence toward not only common average properties but also a common product distribution with oxidative age. This

convergence is seen for all fires studied, as shown in Fig. S7. Whereas the properties and amount of SOA are not investigated

in detail here, *Lim et al.* (2019) observed a correlation between the final mass of SOA and initial VOC concentration, and

showed that this correlation increases with extent of oxidation. This suggests that SOA formed in the oxidation of smoke from

different fuels also tends to converge with OH exposure.

        The extent to which the composition of each BB mixture changes with oxidation, and to which the different mixtures

converge as they age, is examined in more detail in Fig. 4. Differences between any two distributions is quantified in terms of

cosine similarity, the cosine of the angle between two distributions (Ulbrich et al., 2009) (here, the relative concentrations of

all gas-phase species measured by the PTR-MS and I-CIMS), which has values ranging from 0 (completely dissimilar) to 1

(identical). A similar metric has recently been used to examine changes to aerosol mass spectra of evolving biomass-burning

organic aerosol particles (Kodros et al., 2020). Figure 4a shows the cosine similarity between the initial, "fresh" reaction

mixture for each individual fuel and its product distribution as it gradually changes with aging. As the reaction proceeds, each

of the individual fires become less similar to its starting composition (red traces). This contrasts with the blank experiment

(gray trace), for which the composition remains nearly constant (and therefore $\cos\theta \approx 1$) over time.

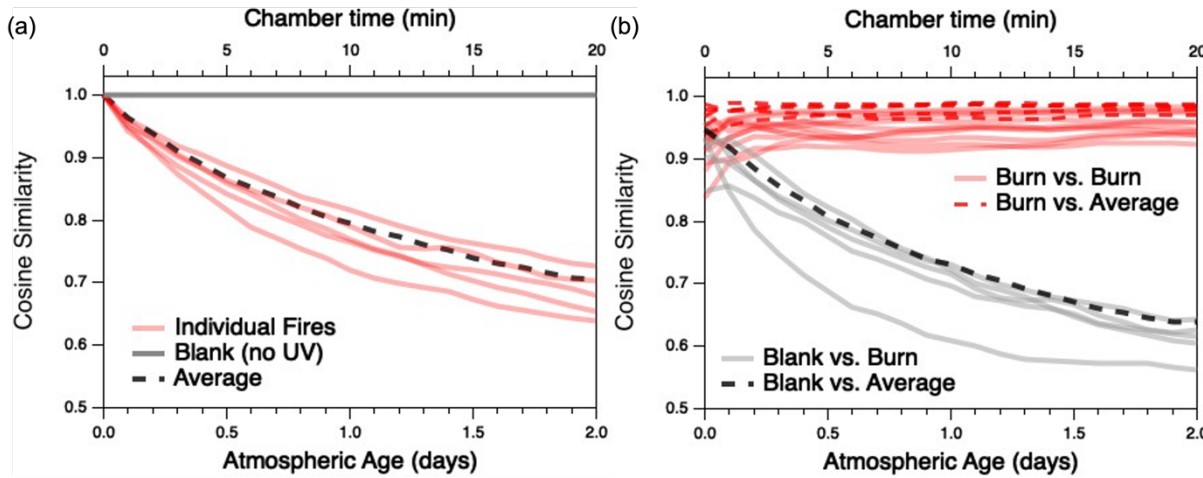





Figure 4. Evolution of cosine similarities between the gas-phase product distributions as a function of atmospheric age; (a) compares the initial distribution of a given fuel with its evolving product distribution, and (b) compares the evolving product distributions at a given age for any two different fuels. Red lines represent comparisons between individual fires; gray lines represent (a) evolution of the mixture in the blank run, or (b) comparison between evolving emissions in the blank run and an oxidation experiment. (Note that comparisons with the blank run are based on PTR-MS measurements only.) Dashed lines represent (a) comparison of the "average fire" (described later in the text) with its initial distribution, or (b) comparisons between individual burns (red) or the blank (black) and the "average fire." Chamber time (top axis) is used for measurements involving the blank run, for which there is negligible oxidative aging.

Figure 4b examines the cosine similarity between different fires as they are oxidatively aged. Although each pair of individual burns (red traces) is generally similar upon introduction into the mini-chamber (initial $\cos\theta$ values range from 0.84-0.96), the product distributions grow more similar as the emissions undergo oxidation (final $\cos\theta$ values range from 0.92-0.98 after 2 days of atmosphere-equivalent oxidation). (Cosine values before and after oxidation are summarized in Table **S1**.) The exception to increasing similarity among all fires is Fire 38 (Ponderosa Pine litter), which becomes slightly less similar to Fires 21 ($\cos\theta$ from 0.93 to 0.92) and 41 ($\cos\theta$ from 0.97 to 0.94), both of which are Lodgepole Pine litter. By contrast, all oxidation systems and the blank run (gray dashed lines) become increasingly dissimilar over time. Overall, these results demonstrate that atmospheric aging not only causes BB emissions to undergo major changes in overall composition, but that these changes result in a general convergence to a product distribution that is relatively similar across all fires, independent of the initial fuel.

This observation of similarly evolving BB product distributions suggests that all burns can be reasonably described by an "average fire," in which normalized gas-phase distributions from each fire are averaged as a function of reaction time. Figure 5a shows the average product distribution (shown as mass spectra) for fresh emissions: in addition to high concentrations of several low-$n_C$ compounds (ethene, acetaldehyde), the reaction mixture contains high levels of unsaturated, high-$n_C$, low-$\overline{OS_C}$ species such as guaiacol, methylfurfural/benzenediol, and monoterpenes. Figure 5b shows the product distribution after two days of atmosphere-equivalent aging. The product distribution is dominated by small ($n_C = 1$-3) VOCs, particularly carbonyls and acids; these include formic acid, formaldehyde, acetaldehyde, acetic acid, and acetone. These distributions are quite similar to those of the individual burns, with cosine similarities of 0.94-0.99 for the fresh emissions and 0.97-0.99 for the aged ones (Fig. 4b and Table S1), indicating these are reasonable representations of all burns studied.

Since the aged distribution (Fig. 5b) is common for all fuels burned, it may serve as a "fingerprint" for aged BB smoke. A list of major compounds in the "average fire" before and after oxidation, as well as the fractional contribution of each, is provided in Table S2. Though several of these species (e.g., formic acid, acetone) are common photooxidation products formed from a range of oxidation systems, the ratios of these products, as well as some of the less common larger species measured, could be potentially useful in ambient studies for identifying atmospherically aged BB emissions. Further, notable subsets of secondary species such as $C_4O_x$ compounds that are generally formed from primary $C_4$-$C_5$ compounds (e.g., furans) have been shown to be good tracers for aged BB emissions (Coggon et al., 2019).

340

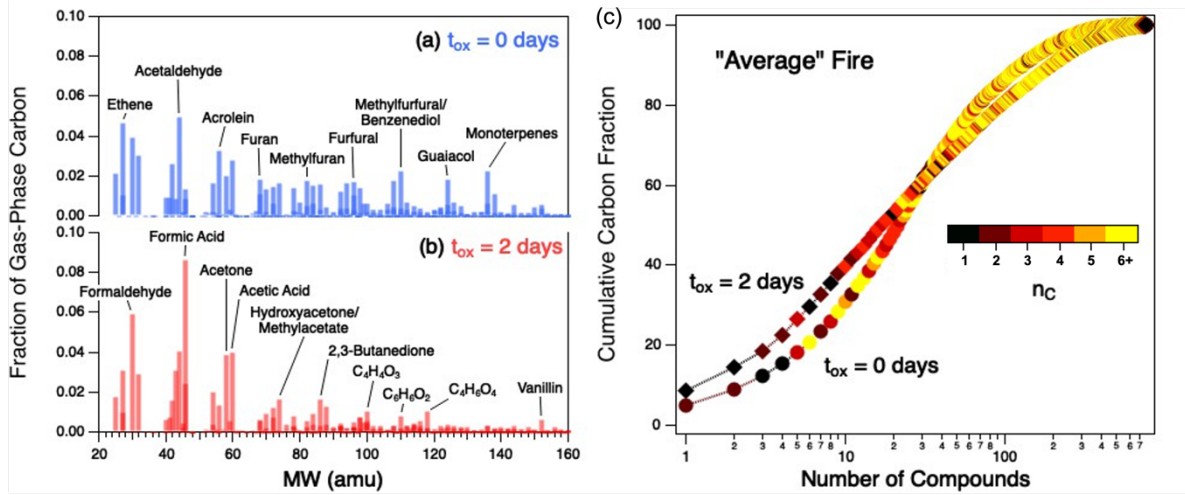

Figure 5. (a-b) Gas-phase product distributions, shown as mass spectra, for the "average fire" for fresh emissions (panel a) and after two days of atmosphere-equivalent oxidation (panel b); (c) cumulative distribution functions (CDFs) showing the percentage of gas-phase carbon for all measured compounds in the "average fire" for fresh emissions (circles) and after two days of atmosphere-equivalent aging (diamonds). Points within individual CDFs are colored by carbon number. CDFs of individual fires can be found in Fig. S8; CDFs colored by other parameters ($\overline{OS_C}$, $c^*$, $\tau_{OH}$) are given in Fig. S9.

The increasing concentration of small ($n_C$ = 1-3) organic species is further illustrated in Fig. 5c, which shows how the chemical complexity of the "average fire," described in terms of cumulative distribution functions (CDFs), evolves with oxidation. (Equivalent CDFs for the individual fires can be found in Fig. S8.) Prior to oxidation (circles), 15 individual compounds of various sizes ($n_C$ = 1-10) cumulatively account for ~40% of gas-phase carbon in the mixture. After oxidation (diamonds), the distribution has evolved such that fewer (~10) compounds, all of which are smaller ($n_C \leq 4$) species, account for the same carbon fraction. Some of these compounds, such as formaldehyde and acetaldehyde, have relatively short atmospheric lifetimes ($\tau_{OH}$ < 1 day); these are commonly formed during oxidation processes, leading to elevated steady-state concentrations. However, many of the compounds that are abundant in the aged mixtures (e.g., formic acid, acetic acid, acetone, and isocyanic acid) tend to be long-lived in the atmosphere ($\tau_{OH}$ extending into 10s of days), which supports the observation of longer average estimated oxidation lifetime of the overall gas-phase mixture (Fig. S4b). This is also observed in our earlier work on -pinene oxidation (Isaacman-VanWertz et al., 2018).

In addition to demonstrating an increased contribution of small VOCs in the aged gas-phase reaction mixture, Fig. 5c also shows that aging increases the overall chemical complexity of the smoke. Specifically, the aged mixture requires substantially more compounds to go from 85% to >97% of the mass than in the fresh mixture (as reflected in the "tail" of the aged mixture, upper right corner of Fig. 5c), indicating an increase in the number of minor organic species in the mixture.





These differences in the product distributions of fresh and aged smoke – aged smoke having comparably fewer products that account for a large fraction of the total carbon, but also a larger number of minor species – provide a broad description of the overall oxidation process of these complex mixtures. The increased abundance of a handful of small and volatile VOCs is driven by fragmentation reactions; this process drives not only the decrease in average gas-phase carbon number (Fig. 2b), but also the increasing similarity between gas-phase product distributions from different burns (Fig. 4b). On the other hand, the increase in the number of minor species (the more pronounced "tail" of the aged CDF in Fig. 5c) is likely the result of functionalization reactions, which involve no changes to carbon number but can involve increases in chemical complexity (Kroll et al., 2011). Such functionalization is also responsible for the simultaneous formation of SOA (Lim et al., 2019).

We have observed a similar bifurcation in previous work, a laboratory study of the products formed in the oxidation of a single VOC (α-pinene) (Isaacman-VanWertz et al., 2018). In that work, oxidation by OH was found to lead to the formation of two pools of products: small volatile oxygenates and SOA components. We see a similar bifurcation here, but with the second pool including not only SOA components but also a large number of functionalized gas-phase species. Moreover, we are able to relate this phenomenon to changes in chemical complexity, as different individual smoke mixtures become increasingly similar to one another with oxidation, largely due to the formation of a few common volatile reaction products. The mixtures still retain a good deal of chemical complexity (the "tail" of Fig. 5c); however, with further oxidation (several more days, not accessed in these experiments) this complexity is also likely to decrease, as such species would likely oxidize further and fragment to form more of the smaller oxygenates. In fact, the observed convergence in composition and the increase in small VOCs may be the early steps towards decreases in chemical complexity with oxidation, something that is expected to happen in all oxidation systems (since, in the absence of other loss processes, the oxidation of organic carbon in the atmosphere will ultimately form one single end-product, $CO_2$). However, to our knowledge this overall decrease in complexity with oxidation has not been described previously, at least for relatively complex systems that involve SOA formation. This may be due to the lack of carbon closure in most previous studies, which precludes a full assessment of chemical complexity, as well as the fact that most laboratory chamber experiments start with a single reduced compound, which will likely lead to an explosion in complexity upon initial oxidation (Isaacman-VanWertz et al., 2018). Here, because the initial mixture was made up of a large number of species, many of which were already oxidized, a decrease in chemical complexity is more likely; studies incorporating multi-phase oxidation and extending oxidation out to still longer timescales are necessary to fully map out the continuing evolution of BB smoke over its full atmospheric lifetime.

## 4 Conclusions

In this study, we oxidized fresh emissions from controlled biomass burning experiments in an environmental chamber, simulating the equivalent of up to two days of OH oxidation in the atmosphere. The oxidative evolution of these emissions





was monitored in both the gas and particle phases by three real-time mass spectrometers, providing insight on the changing concentrations of individual compounds as well as the ensemble properties of the entire mixture. These measurements exhibited an increase in mean carbon oxidation state ($\overline{OS_C}$) and a decrease in molecular size ($n_C$), as well as consistent trends
in mean oxidative lifetime and volatility, across all fuels studied. This is consistent with our previous study of SOA formation, in which SOA formation from a range of different fuels was also found to converge with oxidative aging (Lim et al., 2019).

Despite differences in initial smoke composition, oxidative aging is found to proceed largely independently of fuel type, exhibiting a trend towards a single common gas-phase distribution dominated by a handful of small ($n_C$ = 1-3) oxygenates, as well as a more complex pool of many lower-volatility species. The resulting average product distribution (Fig.
5b) may serve as a "fingerprint" for aged BB emissions and can aid in source apportionment in the far field. This observation also helps to simplify our picture of BB emissions as they are transported through the atmosphere, since the huge variability with fuel type, burn conditions, and other variables might "smooth out" after only 1-2 days of oxidation. This may somewhat reduce the importance of such complexities for the downwind or global effects of biomass burning. However, we note that the present results are based on a relatively small number of experiments, focused on just a handful of fuels that are representative
of the Western U.S., and carried out under a limited range of reaction conditions (fixed temperatures, specific RO$_2$ conditions (Coggon et al., 2019), and oxidation by OH only). Studies examining a wider range of fuels, oxidation conditions, and photochemical ages are needed to assess the extent to which this convergence in composition occurs for all BB emissions, and more generally to provide additional chemical insight into the composition changes associated with the oxidative aging of BB plumes.


**Data Availability**

Data from mini-chamber experiments are available online (https://csl.noaa.gov/groups/csl7/measurements/2016firex/FireLab/DataDownload/).


**Author Contribution**

KJN carried out the analysis and prepared the manuscript along with JHK, with input from all other co-authors. MMC, CYL, ARK, BY, JEK, and KS carried out the experiments and provided the data. JLJ, JdG, CLH, CDC, CW, and JHK conceptualized
and led the study.

**Competing Interests**

The authors declare that they have no competing interests.




## Acknowledgements

This research was supported by the NOAA AC4 program (grant nos. NA16OAR4310112 and NA16OAR4310111).

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
