# Peer review of "Evolution of Organic Carbon in the Laboratory Oxidation of Biomass Burning Emissions"

_Atmospheric Chemistry and Physics, 2022_

## Referee Comment (RC1)

This study quantified and characterized nearly all organic carbon in the laboratory oxidation of biomass burning (BB) emissions through combined mass measurements (AMS, PTR, and I-CIMS). There are several unique elements of this work, including the focus on BB organic species conversion and formation. Generally, the authors provide good context for the work and appropriate references are cited. It is very valuable work. Several specific technical and technical comments are provided below. With minor revisions, the manuscript may be appropriate for publication in ACP.

Specific comments:

1. Line 101 What is the specific concentration or approximate range of fresh plum before aging, and how to define the maximum concentration of particle? The initial concentration of BB plume is very important for the subsequent aging process.

2. The actual OH concentration/exposure in the PFA during the course of each experiment needs to be given if measured.

3. There is a large amount of ozone in the chamber during the experiment. Over the course of the reaction, not only the OH radicals-driven oxidation reaction but also the ozone-driven oxidation reaction. How can the authors prove that OH oxidation dominated in the oxidation mechanism and how much uncertainty will be caused by the existence of high concentration ozone in calculating the equivalent aging time?

4. High-NO (RO2+NO) and low-NO (RO2+HO2) conditions are very important for the aging reaction and oxidation products. The author also stressed this point, but the follow-up manuscript did not pay attention to these chemical regimes, resulting in this part of the content was not well handled.

5. As discussed in the manuscript, only 134 PTR ions and a small subset of CIMS ions were calibrated directly. Can the authors give detailed information about the identified species, such as a table? Another question, I would like to know whether all 763 unique gas-phase species calculated the carbon oxidation state, volatility, and oxidative lifetime, or only identified ions. I wonder if you use different species in the calculation and if it will change the graph.

6. What is the concentration and formation rate of OA measured by AMS over the reaction? The evolution of actual OA concentration cannot be seen in Figure 1. How did the authors calculate the carbon concentration of OA? I can not find the calculation processes in the manuscript. As the author confirmed that the increased abundance of a handful of small VOCs is driven by fragmentation reactions, however, fragmentation reactions generally lead to the reduction of OA (doi.org/10.5194/acp-11-3303-2011, doi.org/10.1002/2014JD022563). Does this conflict with the growth of OA?

7. What is the reason for the decrease in carbon concentration of AMS, I-CIMS, and PTR in the

initial stage in Figure 1?

8. Are the species in Table 2 only measured by PTR or were detected by both instruments?Can you mark which instrument detected these species respectively?

9. The authors emphatically analyzed the Fire 25 and Fire 26 experiments. The two groups of experiments used the same fuel and got similar commons. However, the MCE and moisture content of these two groups of fuels are significantly different. Many previous field and laboratory studies have emphasized the aging process of high MCE (flaming) and low MCE (smoldering) and found that there are great differences, including the SOA formation, the oxidation state of OA, and gaseous oxidation products (doi:10.1029/2021JD034534), change of optical properties (10.1021/acs.est.0c07569), and the influence of aerosol emissions from wildfires driven by MCE (doi.org/10.1021/acs.est.6b01617). Will this factor probably influence your results, which may be worth some discussion and explanation?

Technical corrections:

Line 34, the comparison of mass spectra
Line 127, in the measurement
Line 136, delete the second "and"
Line 144, due to the unavailability of standards,
Line 301, each of the individual fires becomes...

---

## Author Comment (AC1)

Commenter #1
This study quantified and characterized nearly all organic carbon in the laboratory oxidation of biomass burning (BB) emissions through combined mass measurements (AMS, PTR, and I-CIMS). There are several unique elements of this work, including the focus on BB organic species conversion and formation. Generally, the authors provide good context for the work and appropriate references are cited. It is very valuable work. Several specific technical and technical comments are provided below. With minor revisions, the manuscript may be appropriate for publication in ACP.

Thank you for your comments! We have responded to them below in blue.

Specific comments:

1. Line 101 What is the specific concentration or approximate range of fresh plum before aging, and how to define the maximum concentration of particle? The initial concentration of BB plume is very important for the subsequent aging process.

   This is a good suggestion; we have updated the text to include a range of NMOG and particulate concentrations prior to the initiation of oxidation. The updated text starting at line 101 now reads as follows:

   "Filling continued until particle concentrations reached a maximum, resulting in pre-oxidation NMOG concentrations between ~100-300 ppb and particle concentrations between ~10-30 µg/m$^3$ for the experiments studied here (Lim et al., 2019)."

2. The actual OH concentration/exposure in the PFA during the course of each experiment needs to be given if measured.

   The average concentration of OH was added to the paper following the discussion of OH generation of line 107. The sentence starting on line 104 now reads as follows:

   "After allowing time for mixing, one 40 W UVC lamp (narrow peak emission at 254 nm, Ultra-Violet Products, Inc.) was turned on to generate OH (via photolysis of ozone, followed by reaction of O($^1$D) with H$_2$O, as well as the photolysis of other precursors in the smoke such as HONO) which initiated oxidation (average [OH] ≈ 2 x 10$^8$ molec cm$^{-3}$) (Lim et al., 2019)."

3. There is a large amount of ozone in the chamber during the experiment. Over the course of the reaction, not only the OH radicals-driven oxidation reaction but also the ozone-driven oxidation reaction. How can the authors prove that OH oxidation dominated in the oxidation mechanism and how much uncertainty will be caused by the existence of high concentration ozone in calculating the equivalent aging time?

   There may be a small contribution of O$_3$ in the chamber, but Koss et al. 2018 shows that the reaction mixture does not contain an abundance of double bonds, which limits the contribution of ozone reactivity. Further, the concentration of O$_3$ is roughly constant

across different burns but these BB mixtures are still observed to converge, meaning that the contribution of $O_3$ to overall reactivity does not meaningfully impact our findings. In order to account for the role of $O_3$ in the reaction mixture, the following sentence was added to line 113:

"Oxidation by $O_3$ is unlikely to play a major role in the observed oxidation, as only a small fraction of the NMOGs have aliphatic C=C double bonds (Koss et al., 2018)."

4. High-NO (RO2+NO) and low-NO (RO2+HO2) conditions are very important for the aging reaction and oxidation products. The author also stressed this point, but the follow-up manuscript did not pay attention to these chemical regimes, resulting in this part of the content was not well handled.

This is a good point, and our initial discussion on line 115 warrants a follow-up. The impacts on $NO_x$ concentrations and the difference between the High- and Low-NO conditions is handled in greater detail in Coggon et al., 2019, as referenced in the manuscript most relevantly on line 116. That paper analyzed fires with substantially different VOC/$NO_x$ ratios, and their Figure S14 demonstrates that oxidation chemistry is dominated by $RO_2 + HO_2$ regardless of initial VOC/$NO_x$ ratio. These observations are also consistent with what is seen in the field: Xu et al., 2021 and Robinson et al., 2021 show that the RO2 + HO2 chemistry becomes dominant within 1-3 hrs of atmospheric aging for plumes emitted during daytime hours.

Overall, while there is a wide range of experimental parameters that we do not examine in great detail, we observe a trend towards convergence in the BB mixtures that is independent of a range of initial factors. As such, we have added following text on line 350 after we discuss the observed convergence and trends towards an "average fire:"

"While the full range of experimental parameters (e.g., initial VOC/$NO_x$ ratio, MCE, moisture content) is not explored in this work, there is no obvious dependence on these factors on the trend towards convergence."

5. As discussed in the manuscript, only 134 PTR ions and a small subset of CIMS ions were calibrated directly. Can the authors give detailed information about the identified species, such as a table? Another question, I would like to know whether all 763 unique gas-phase species calculated the carbon oxidation state, volatility, and oxidative lifetime, or only identified ions. I wonder if you use different species in the calculation and if it will change the graph.

In response to the first point: Koss et al., 2018 provides a complete list of compounds identified by the PTR, so such a list is not recreated here. Because we also used I-CIMS data, we have taken your suggestion to include a list of the identified I-CIMS species, along with some of their molecular parameters, in the SI. So we have added Table S1 to the supplement (and updated the numbering of the other tables in the SI/manuscript), and have reflected this change in the paper on line 149 as follows:

"Due to the unavailability of standards, I-CIMS signals could not be meaningfully calibrated into concentrations for the vast majority of ions; only a small subset of ions was calibrated directly (see Table S1), allowing for the determination of a maximum instrument sensitivity (~8000 ncps/ppb)."

In response to the second question: the paragraph beginning on line 164 describes in detail that carbon oxidation state, volatility, and oxidative lifetime are either directly implemented in the case of known species, or calculated based on molecular formulae for unknown species. In other words, all detected ions are used in these ensemble measurements.

6. What is the concentration and formation rate of OA measured by AMS over the reaction? The evolution of actual OA concentration cannot be seen in Figure 1. How did the authors calculate the carbon concentration of OA? I can not find the calculation processes in the manuscript. As the author confirmed that the increased abundance of a handful of small VOCs is driven by fragmentation reactions, however, fragmentation reactions generally lead to the reduction of OA (doi.org/10.5194/acp-11-3303-2011, doi.org/10.1002/2014JD022563). Does this conflict with the growth of OA?

While this paper does discuss the evolution of OA in these aging BB mixtures, the growth of OA is addressed primarily in terms of changes to ensemble measurements. A full treatment of SOA formation and aging would be redundant for this study in that it is provided in great detail in Lim et al., 2019, which is referenced multiple times within this paper.

As to the reviewer's second point, the observation that many small VOCs are formed by fragmentation is not in conflict with the growth of OA, as discussed in the paragraph starting on line 395. Specifically, we see the complex reaction mixture follow two dominant pathways: fragmentation resulting in smaller gas-phase species and simultaneous functionalization resulting in more partitioning to the particle phase, the latter of which is discussed in Lim et al., 2019 as being the primary driver of SOA aging (as opposed to heterogeneous oxidation). We feel that this discussion starting on line 397 sufficiently explains this phenomenon:

"The increased abundance of a handful of small and volatile VOCs is driven by fragmentation reactions; this process drives not only the decrease in average gas-phase carbon number (Fig. 2b), but also the increasing similarity between gas-phase product distributions from different burns (Fig. 4b). On the other hand, the increase in the number of minor species (the more pronounced "tail" of the aged CDF in Fig. 5c) is likely the result of functionalization reactions, which involve no changes to carbon number but can involve increases in chemical complexity (Kroll et al., 2011). Such functionalization is also responsible for the simultaneous formation of SOA (Lim et al., 2019)."

7. What is the reason for the decrease in carbon concentration of AMS, I-CIMS, and PTR in the initial stage in Figure 1?

We have seen this phenomenon in several other studies, and our observations suggest that it may be due to an initial partitioning of some oxidation products to chamber walls and/or instrument inlets. While we do not have conclusive evidence of this, we updated the text on line 234 as follows:

"In all cases (Figs. 1 and S1), after an initial dip in carbon concentration (which may be a result of short-term partitioning of some oxidation products to the chamber walls/instrument inlets), there is a clear overall increase in total measured organic carbon over the course of the reaction; this increase is not large (typically 15-35%)."

8. Are the species in Table 2 only measured by PTR or were detected by both instruments? Can you mark which instrument detected these species respectively?

All gas-phase data included in this paper includes both PTR and I-CIMS measurements. In this table, all of the top species that contribute $\geq 1.0\%$ of carbon to the mixture happen to have been measured by the PTR, so we have added the following line of text to the caption under Table S2, on line 129 of the SI:

"While gas-phase data in this work are measured by both the PTR and I-CIMS, all compounds contributing $\geq 1.0\%$ of carbon concentration to the mixture were detected by the PTR."

9. The authors emphatically analyzed the Fire 25 and Fire 26 experiments. The two groups of experiments used the same fuel and got similar commons. However, the MCE and moisture content of these two groups of fuels are significantly different. Many previous field and laboratory studies have emphasized the aging process of high MCE (flaming) and low MCE (smoldering) and found that there are great differences, including the SOA formation, the oxidation state of OA, and gaseous oxidation products (doi:10.1029/2021JD034534), change of optical properties (10.1021/acs.est.0c07569), and the influence of aerosol emissions from wildfires driven by MCE (doi.org/10.1021/acs.est.6b01617). Will this factor probably influence your results, which may be worth some discussion and explanation?

Thanks for this comment. The effects of MCE and moisture content, among other factors, are important considerations in BB aging, as observed in the papers cited by the Reviewer. While the fuels in this paper do indeed span a range of initial conditions (MCE, moisture content, fuel composition, etc.), we find that our general observations of how the reaction mixtures converge into two dominant channels (an increase in small VOCs from fragmentation and SOA from functionalization) is consistent across all the studied fuels. We have updated the text on line 350 with the following text, as discussed in comment #4 above.

"While the full range of experimental parameters (e.g., initial VOC/NO$_x$ ratio, MCE, moisture content) is not explored in this work, there is no obvious dependence on these factors on the trend towards convergence."

10. Technical corrections:

    a.  Line 34, the comparison of mass spectra

       We have made this correction. The updated text on Line 34 now reads as follows:

       "In addition, the comparison of mass spectra from the different fuels indicates that the oxidative evolution of BB emissions proceeds largely independent of fuel type, with different fresh smoke mixtures ultimately converging into a common, aged distribution of gas-phase compounds."

    b.  Line 127, in the measurement

       We have made this correction. The updated text on Line 132 now reads as follows:

       "The overall absolute uncertainty in the measurement of organic particulate mass is estimated to be 38%, as discussed elsewhere (Bahreini et al., 2009)."

    c.  Line 136, delete the second "and"

       We have made this correction. The updated text on Line 141 now reads as follows:

       "Of the 291 unique ions detected by the PTR, 134 were identified, and the remaining 157 are unidentified…"

    d.  Line 144, due to the unavailability of standards,

       We have made this correction. The updated text on Line 149 now reads as follows:

       "Due to the unavailability of standards, I-CIMS signals could not be meaningfully calibrated into concentrations for the vast majority of ions;"

    e.  Line 301, each of the individual fires becomes…

       We have made this correction. The updated text on Line 325 now reads as follows:

       "As the reaction proceeds, each of the individual fires becomes less similar to its starting composition (red traces)."

Commenter #2

The authors detail the evolution of reactive organic carbon from biomass burning emissions at the FIREX FireLab campaign. Approximate carbon closure was achieved using three instruments, following a similar analysis from previous work. Generally, the gas phase reactive carbon moves to smaller carbon number and higher carbon oxidation state. The spectra of compounds in aged smoke all looked rather similar, regardless of which fuel was burned. This

work represents an important step towards understanding the chemistry of aged smoke, which has air quality impacts all around the world. The paper is very well written. I recommend for publication after addressing my comments below.

Thank you for your comments! We have responded to them below in blue.

Specific comments:

1. Line 104: What is the estimated OH concentration in the chamber?

   The average concentration of OH was added to the paper following the discussion of OH generation of line 107. The sentence starting on line 104 now reads as follows:

   "After allowing time for mixing, one 40 W UVC lamp (narrow peak emission at 254 nm, Ultra-Violet Products, Inc.) was turned on to generate OH (via photolysis of ozone, followed by reaction of $O(^1D)$ with $H_2O$, as well as the photolysis of other precursors in the smoke such as HONO) which initiated oxidation (average $[OH] \approx 2 \times 10^8$ molec cm$^{-3}$)."

2. Line 146: Can you give more details about how this max sensitivity of 8000 ncps/ppt was estimated? E.g., which compound(s) were calibrated with that sensitivity? That number seems way larger than expected, and way larger than the values of 300 ncps/ppt (theoretical) or 75 ncps/ppt (empirical) used in the similar analysis of Isaacman-VanWertz et al. 2018. Can you specifically say why the value you use is so much larger in this paper? Was a different ion-molecule reactor (IMR) used on the CIMS that had a much longer residence time or some other change like that?

   Thank you for catching this error! The maximum sensitivity should be written as 8000 ncps/ppb, not ncps/ppt. We have updated this on line 149:

   "Due to the unavailability of standards, I-CIMS signals could not be meaningfully calibrated into concentrations for the vast majority of ions; only a small subset of ions was calibrated directly, allowing for the determination of a maximum instrument sensitivity (~8000 ncps/ppb)."

3. Line 160: The I-CIMS probably measures a lot of peroxides though. Uncertainty analysis?

   As mentioned in line 175, unambiguously detecting peroxide compounds (and peroxyacyl nitrates in particular) is not feasible with the suite of instruments used in this experiment. While there are likely peroxides in this reaction mixture that may be detected by the I-CIMS, the uncertainty associated with the I-CIMS is already large enough such that incorporating further uncertainty analysis into this subset of compounds would likely not yield any greater clarification on their role in this reaction mixture. As such, we elect to limit our discussion of peroxides to the discussion in line 175 about PANs.

4. Line 192: The "(1  )" looks like a typo?

Thanks for pointing this out; it seems there was a formatting error. This empty symbol has been replaced with a sigma to represent a standard deviation. The updated text on line 196 now reads as follows:

"Error bars represent the uncertainty ($1\sigma$) of the total concentrations before and after aging, for individual instruments as well for the overall reaction mixture."

5. Line 237: In addition to fragmentation reactions, the decreasing n_c could be due to partitioning of larger n_c compounds into the particle phase after functionalization, right?

This is an excellent point. The smaller average carbon size in the gas phase could be attributed to partitioning of larger-$n_C$ species in the particle phase, and it is worth including this idea in the paper. The sentence on line 258 of the paper has been updated to include this comment as follows:

"All burns characterized here, independent of fuel, exhibit the same overall trends, in which OH oxidation leads to increases in the oxidation state of the organic carbon (Fig. 2a), as well as decreases in mean carbon number (Fig. 2b); this latter observation is primarily attributable to fragmentation reactions, and may also be influenced by the partitioning of higher-$n_c$ species into the particle phase, as discussed below."

This comment is referenced again on line 397, which has likewise been updated as follows:

"The increased abundance of a handful of small and volatile VOCs is driven by fragmentation reactions and the loss of higher-$n_C$ gas-phase species that have partitioned into the particle phase;"

6. Line 245: Do you have a possible explanation for why the volatility briefly increases before decreasing gradually, shown in Fig. S4a and S5? Please add it to the text, that will help the reader to understand why you are showing this data.

This is a good question that warrants further discussion. As such, line 66 of the SI has been updated to include the following text:

"This observation is likely driven by the fact that the average gas-phase carbon number, $n_C$, rapidly decreases during the first ~0.5 days of atmospheric oxidation, whereas the average gas-phase oxidation state, $OS_C$, more gradually increases throughout the course of the reaction. Combined, these phenomena result in a quick rise in volatility (as many low-$n_C$ compounds are formed in a short interval) followed by a more gradual decay (as the gas-phase mixture is steadily oxidized while the carbon number remains relatively constant)."

7. Line 325: Are the mass spectra in Fig. 5a+b combined PTR and I-CIMS data? Could be good to say that. I guess the answer would also apply to all of the figures. It might be interesting to make a version of Fig. 5a+b, or better yet Fig. 3, for the SI where you differentiate which

compounds were measured by which instrument. This is not necessary for drawing your conclusions in this work, so feel free to ignore this comment, but it might be an interesting bit of extra information to show how the PTR and I-CIMS measure complementary parts of the spectrum of compounds.

> All of the gas-phase data in this study includes both PTR and I-CIMS measurements. We feel that this is addressed in the methods section, particularly in line 162: "Overall, 50 overlapping traces were removed, leading to a total of 763 unique gas-phase species." As such, we do not see the need to indicate that both instruments were used in this figure.
>
> We appreciate the latter part of the comment about recreating these figures with differentiating traces by instrument, as that could provide useful info for measuring complex datasets. However, we believe this would make these figures a bit too busy and outside of the scope/goals of this particular manuscript, so we will leave them as they are.

8. Line 338: Phenolics etc. can also fragment to produce C4Ox products, so maybe make this statement a little more general that C4Ox compounds are formed from C4+ precursors including furans or larger precursors?

> This is a good suggestion. The text on Line 366 has been updated to include these other classes of compounds as follows:
>
> "Further, notable subsets of secondary species such as $C_4O_x$ compounds that are generally formed from primary $C_4$-$C_5$ compounds (e.g., larger precursors such as furans and phenolic species) have been shown to be good tracers for aged BB emissions (Coggon et al., 2019)."

9. Line 348: I think it would be useful for you to explain how you make a cumulative distribution function (CDF) a bit more. I am unfamiliar with them, and it took me a long time to understand. I guess the compounds are added starting with the highest concentrations first?

> To clarify how CDFs are constructed, we have added the following line of text to line 118 of the SI:
>
> "These cumulative distribution functions (CDFs) represent the total fraction of gas-phase carbon in the system as a function of number of compounds, and are arranged such that compounds are added from highest to lowest concentrations within the reaction mixture."

10. Line 366: Again, could loss of higher n_c compounds to the particle phase through functionalization and condensation contribute to this result of decreasing average n_c and higher relative importance of small VOCs? If you find that is a minor contribution, can you say so with any evidence?

> We agree with this point; this comment is addressed in comment #5 above.

11. Line 366 part two: Could heterogeneous oxidation of aerosols lead to evaporation of small n_c oxidation products? Do you expect any meaningful heterogeneous oxidation at your high OH concentrations?

> This is a good point, and something that we considered during our analysis of the data. The text beginning on line 316 notes that the growth of SOA is discussed in detail in Lim et al., 2019, and specifically concludes that continual oxidation of the gas-phase mixture is responsible for the growth in SOA. Lim's study measures the effects of heterogenous oxidation of SOA and ultimately concludes that it has a small impact on the growth/aging of SOA. It still might play a role in gas-phase composition, possibly leading to increased concentrations of low-$n_C$ species; but this would be a result of fragmentation reactions, which are specifically mentioned in line 398.